# Internet of Things for Mental Health: Open Issues in Data Acquisition, Self-Organization, Service Level Agreement, and Identity Management

**DOI:** 10.3390/ijerph18031327

**Published:** 2021-02-01

**Authors:** Leonardo J. Gutierrez, Kashif Rabbani, Oluwashina Joseph Ajayi, Samson Kahsay Gebresilassie, Joseph Rafferty, Luis A. Castro, Oresti Banos

**Affiliations:** 1Sonora Institute of Technology (ITSON), Ciudad Obregon 85130, Mexico; leonardo.gutierrez@potros.itson.edu.mx; 2School of Computing, Ulster University, Newtownabbey BT37 0QB, UK; rabbani-k@ulster.ac.uk (K.R.); ajayi-o2@ulster.ac.uk (O.J.A.); gebresilassie-s@ulster.ac.uk (S.K.G.); j.rafferty@ulster.ac.uk (J.R.); 3CITIC-UGR Research Center, University of Granada, 18014 Granada, Spain; oresti@ugr.es

**Keywords:** mental health, Internet of Things (IoT), security, self-organization, service level agreement

## Abstract

The increase of mental illness cases around the world can be described as an urgent and serious global health threat. Around 500 million people suffer from mental disorders, among which depression, schizophrenia, and dementia are the most prevalent. Revolutionary technological paradigms such as the Internet of Things (IoT) provide us with new capabilities to detect, assess, and care for patients early. This paper comprehensively survey works done at the intersection between IoT and mental health disorders. We evaluate multiple computational platforms, methods and devices, as well as study results and potential open issues for the effective use of IoT systems in mental health. We particularly elaborate on relevant open challenges in the use of existing IoT solutions for mental health care, which can be relevant given the potential impairments in some mental health patients such as data acquisition issues, lack of self-organization of devices and service level agreement, and security, privacy and consent issues, among others. We aim at opening the conversation for future research in this rather emerging area by outlining possible new paths based on the results and conclusions of this work.

## 1. Introduction

Mental illness has been described as one of the most serious global health problems, characterized by a growing number of patients suffering from depression, anxiety, and other disorders [1]. Mental health is integral to overall wellbeing and has been defined by the World Health Organization (WHO) as “a state of wellbeing in which every individual realizes his or her own potential, can cope with the normal stresses of life, can work productively and fruitfully, and is able to contribute to her or his community” [2]. According to the WHO, currently, more than 500 million people suffer from mental health disorders, among which depression, schizophrenia, and dementia are especially prevalent worldwide [3]. As predicted and exacerbated by the COVID-19 pandemic, mental conditions such as depression have become a leading cause of disability [4].

With the advent of the Internet of Things (IoT), and related paradigms such as ubiquitous computing, the opportunities for developing technologies to assist professionals and patients with mental conditions have significantly increased. However, despite recent advances in IoT-based assistive technologies [5,6,7], not much research has been devoted to mental health care applications. This is why it is important to study how IoT can be of help in identifying, assessing, treating, and potentially alleviating symptoms associated with mental disorders.

The paradigm posed by IoT technologies and systems combines physical and virtual elements that span across an ever-increasing range of elements such as electronic components, sensors, actuators, and software to provide data-driven products and services [8,9]. With the rapid rise in the development of IoT devices [10], the IoT paradigm promises to connect everything across all industries enabling near constant and seamless access to information and suggestions that affect everyday decisions [11]. IoT has positively impacted many sectors including agriculture [12], supply chain [13], smart home [14], smart cities [15], health care [16], and others. IoT devices are commonly used for monitoring and collecting a wide range of data, ranging from physiological signals all the way through social and behavioral ones [17,18]. These data are typically generated through on-device sensors such as proximity sensors, ambient light sensors, accelerometers, gyroscopes, magnetometers, ambient sound sensors, barometers, temperature, and humidity sensors, to name a few.

Among the industries that IoT is expected to have a great impact is healthcare. In this regard, IoT has been contributing through innovative solutions in monitoring, wellbeing interventions, self-quantification, and information services contributing to improved patient quality of life and wellbeing [19,20]. However, there are still many opportunities to advance research in this area, particularly in mental health. IoT technologies have a great potential in mental health for diagnoses, treatment, and care due to the increased ability to collect real-time data indicating patterns of activity and behavior of people [21]. In addition, these solutions enable patients to remotely interact with health care professionals, facilitating discussion about patients’ perceptions, their health and conditions for more effective monitoring and assessment [22]. Still, the adoption of IoT technologies for mental health has been slow, which can be partially ascribed to the misconceptions that exist wherein patients may pose an increased risk to themselves and others [23].

In this work, we present a survey aimed at showing how IoT can support mental health. We used the Diagnostic and Statistical Manual of Mental Disorders 5 (DSM 5) [24], an authoritative reference defining and categorizing mental health disorders, in order to carefully determine how current IoT systems and services can be of use for diagnosis, treatment, follow-up, and enhancement of mental health. There is, however, an ongoing discussion about the equivocal nature of some of the symptoms in the DSM 5, which can lead to overlaps certain disorders [25,26]. In this work, we aim at reconciling studies with the terminology and expert classification (i.e., based on the DSM 5), as opposed to employing generalized non-expert criteria, typically found in technical works.

There are other surveys that focus on mental health and technologies. Some of them have focused on analyzing the efficacy of smartphone apps for mental health [27], or guiding the development of those apps [28]. Also, other reviews have explored the use of chatbots in mobile health care [29] as well as their most common features [30]. Moreover, the use of social networking sites for mental health such as Facebook has also been explored [31]. Finally, other surveys have focused on showing the impact of ubiquitous computing research and machine learning techniques and methods used in mental health computing [32,33]. In this work, we focus on surveying open issues in data acquisition, self-organization, service level agreement, and identity management during mental health interventions. In particular, we present a collection of IoT platforms, devices, findings, and challenges in published works. From this work, a number of relevant challenges are identified that relate to patients’ privacy and confidentiality enhanced by service level agreement between the patient and the IoT systems; security challenges such as device identity issues and data security; organization of IoT systems; and the relationship between the number of devices in an IoT platform and its effectiveness in acquiring all relevant data needed for an effective diagnosis. Also, we used the DSM 5 to categorize each of the mental disorders reported in the literature, which, to the best of our knowledge, had not been done in previous related studies.

The rest of the manuscript is organized as follows: Section 2 shows the methodology used in the selection of studies for this survey, Section 3 introduces the most prevalent mental health disorders; Section 4 presents the IoT technologies that could be used to sense the behavioral, physiological and social signals related to early detection of mental health disorders. Section 5 presents the background and current state of the art in IoT based mental health systems. Section 6 presents the challenges identified through this survey and identifies limitations regarding data acquisition, self-organization, service level agreement, security and identity management in the use of IoT for mental health. Section 7 concludes the paper, summarizing findings and providing potential directions for future work regarding the challenges discussed in Section 6.

## 2. Methods

We performed a survey of the literature on IoT-based mental healthcare and wellbeing systems with a particular focus on prominent aspects such as data acquisition, self-organization, service level agreement, security, and identity management.

### 2.1. Search Strategy

#### 2.1.1. Sources

We performed searches in scientific databases and commercial academic search engines for two relevant domains, namely “Computing and Technologies” (ACM digital library, Mendeley, Google Scholar) and “Medicine” (Elsevier eLibrary, Google Scholar). Cross-domain searches were performed in the aforementioned databases and digital libraries by using the terms shown next.

#### 2.1.2. Terms

The search was undertaken using the following terms: “Mental Health”, “Mental Disorders”, “Wellbeing”, “Internet of Things”, “Internet of Things based Systems”, “Data Acquisition”, “Service Level Agreement”, “Self-Organization”, “Security”, “Identity Management”. The following search queries were used: “Mental Health” and (“Internet of Things” or “Internet of Things based Systems”); “Mental Disorders” and (“Internet of Things” or “Internet of Things based Systems”); “Self-Organization” or “Security” or “Identity Management” or “Service Level Agreement” and (“Mental Health” or “Mental Disorders”); and, “Internet of Things based Systems” or “Internet of Things”.

### 2.2. Study Eligibility Criteria

Studies were selected if they used IoT technologies for the collection and processing of data and they focused on mental health and wellbeing monitoring applications. Smartphones, wearables and ambient sensors were considered as IoT technologies, whereas mental health related disorders were extracted from the DSM 5. However, we included only manuscripts on the most prevalent mental health conditions, namely bipolar disorders, depression, schizophrenia, and stress-related disorders. We included only works that were published in the English-language from 2010 to 2020 (inclusive).

## 3. Prevalent Mental Health Disorders: Assessment and Challenges

Certain mental health disorders are better suited for applying IoT technologies, specifically those disorders involving observable changes in behavior or that do not impair the patients’ judgement. In this section, we discuss the leading mental disorders that have been addressed from a technical perspective such as bipolar disorders, depression, schizophrenia, and stress-related disorders. The order in which they are presented is by no means related to their importance, prevalence, or any particular criterion. This section mainly focuses on providing an overview of symptoms, diagnosis, and treatment. We also present studies that have addressed these disorders, which are thoroughly described in Section 5.

First, bipolar disorders, also referred to as manic-depressive illness, are characterized by atypical mood changes. These changes are defined by episodes of significantly elevated energy levels or arousal (mania), contrasting with episodes of extremely low mood (depression) [24]. When not treated adequately, these disorders can lead to serious consequences such as suicide and self-harm [34]. Typical treatments include Cognitive-Behavioral Therapy (CBT) [35,36], thus the use of technology to monitor changes in behavior can be of great help for health care professionals and patients [37,38,39,40].

Also, with increased interest, depressive disorders, also known as major depressive disorder or unipolar depression, are characterized by constant low mood during most of an individual’s life. Symptoms include loss of interest in day-to-day activities, significant weight changes, reduction of mobility, constant fatigue, difficulty to concentrate, and feelings of worthlessness. These symptoms can lead to suicidal thoughts and increase the risk of self-harm [24]. Current instruments for professional assessment include psychometric inventories such as the Patient Health Questionnaire-9 (PHQ-9), which can be used to diagnose depressive disorders and their severity [41]. Still, many of these inventories are applied by therapists and are based on the patients’ self-report. Apart from digitalization and automated analysis of PHQ-9, not many IoT technological solutions have been developed to help treat and support individuals with this condition [42,43,44].

In addition, schizophrenia spectrum disorder, also referred to as schizophrenia spectrum and other psychotic disorders, is characterized by five psychopathological domains incorporating hallucinations, delusions, disorganized thought, abnormal motor behavior, and negative symptoms, which have been used to assess this type of disorder [45]. Some IoT driven systems have been developed to monitor patients with this condition, providing a non-invasive solution for treating this disorder [46,47,48].

Finally, stress-related disorders, defined as trauma and stressor-related disorders, such as posttraumatic stress disorder (PTSD) and acute stress disorder (ASD) are characterized by symptoms arising from exposure to an extreme stressor. Extreme stressors may include the threat of death, incidents causing injuries or sexual violence. Symptoms may emerge include flashbacks, nightmares, and hypervigilance [49]. These disorders, if not treated correctly, can result in negative consequences such as marital instability, self-neglect, and unemployment [50].

Clinical approaches to tackle the aforementioned disorders include pharmacological (e.g., mainly psychiatric treatments) and non-pharmacological interventions (e.g., mainly psychological interventions). Approaches driven by IoT technologies have remained largely unexplored. However, some studies have shown their potential for detecting, assessing or dealing with these disorders [51]. In addition to directly treating these illnesses or disorders, changes in behavior can be monitored to quantify progression, deterioration, or effectiveness of treatment [52].

## 4. Measuring Mental Health Related Data

Change in mental illness can be reflected by analyzing data linked to behavioral, psychological, and social signals [53]. These data can be collected using IoT related technologies, which are next described:

### 4.1. Behavioral Data

Behavioral features are the observable part of an individual’s actions and intentions. Data encapsulating these features represent characteristics that can be monitored such as location, speech features, patient activity, and the interaction with several technologies such as smartphones or smartwatches. Within the domain of mental health, studies have shown that metrics generated from these data can indicate an individual’s mental state. For instance, there has been a commonly accepted link between depression and the sedentary activity [54]. An individual’s location and physical mobility can be inferred using satellite-based tracking, accelerometer sensors, and trilateration using radio signals such as Wi-Fi. Some studies have used such data to determine mobility features which subsequently correlated with severe depression [43]. Also, speech patterns can also be traced by using the embedded phone and/or smartwatch microphones. Voice features, such as frequency range, rate of speech, or the volume can be a significant indicator of individuals suffering from a mental disorder, specifically depression [55].

The usage of everyday technologies such as smartphones can help determine the mental state of a person. Studies have shown that the frequency of using a smartphone changes across different states of people diagnosed with bipolar disorder [56]. Another class of behavioral data type representing activity level can be traced using some of the more commonly adopted sensors such as accelerometers and a gyroscope. Such data can be an indicator of mental health state. For instance, patients with schizophrenia tend to decrease their level of physical activity when the severity of their illness arises [57].

### 4.2. Physiological Data

Physiological data such as facial expression, heart rate, eye movement, electrodermal activity (EDA) can be important markers for assessing mental health conditions [58]. Recognizing facial expressions or changes in the movement of the eyes can be determined by cameras placed in a mobile phone, which can provide significant cues related to the mental state of the patient [59]. Also, the study of eye movement and blinking between depressed and non-depressed individuals showed that this discrimination is feasible [60].

Furthermore, heart rate can also be used to determine an individual’s mental state. Some studies have determined that the heart rate variability has a link with mortality in patients with mental disorders, specifically the ones suffering from depression [61]. Finally, higher levels of EDA, the levels of electrical properties of the skin, can be traced by using wearable sensors and can develop in negative symptoms and poor functional outcomes in patients with schizophrenia [62].

### 4.3. Social Data

Social features can be regarded as the level of engagement in a person’s interactions with others, which can be used as an indicator of mental wellness [53]). Social interactions can be traced using Bluetooth, Wi-Fi or the microphone to recognize location and proximity between individuals [63]. These technologies have been used to capture movement and interaction between patients with schizophrenia, providing information regarding the way patients interact with each other as well as usability and acceptability concerns [64]. Also, the use of social media such as Facebook, Instagram or Twitter can be used to determine the patients’ mental state. For instance, the use of Instagram photos to detect features that can predict depression symptoms between their users [65].

Overall, the analysis of data associated with behavioral, psychological, and social signals can provide significant cues regarding internal states of individuals. Some of these data can be effectively collected using existing technologies, and can be of great help to supplement self-report of patients for providing better diagnoses and personalized treatment by health care professionals.

## 5. IoT Systems for Mental Health and Wellbeing

Some works in the literature have been already contributing to efforts at the intersection between IoT and mental health. In this section, we present and discuss the IoT systems that have been aiming at providing technological solutions to some of the most prevalent mental health conditions.

### 5.1. Bipolar Disorders

Due to the nature of IoT technologies, new possibilities for collecting physiological, behavioral, and social data from individuals are being developed. A concrete example of this is MONARCA, a personal mobile monitoring system aimed at people with bipolar disorders. The system facilitates the constant monitoring of patients’ mood, as well as quick access to the patients’ information by doctors or family members [52], though it does not offer personalization capabilities in terms of which data to collect and how to visualize it. Another approximation for treating manic-depressive disorder (Bipolar I) is the use of smartphones for the continuous monitoring of peoples’ behavioral patterns such as location and mobility, level of activity, and social interaction which represent potential indicators that could detonate a change in mental state [38]. Nonetheless, due to the nature of the patients’ disorders and the ability to turn on or off the measurement based on their desire, they presented difficulties in the patients’ willingness to participate during the study trials [38]. In a similar fashion, Psyche stands as a continuous monitoring system to treat and predict bipolar disorder episodes, integrating both physiological and behavioral sensing, providing the user with an interface for personal and disease-wise data management [40], although the use of textile sensors may be used as a basis for future studies, less invasive ones are yet to be developed. Studies related to bipolar disorders and smartphone usage (e.g., phone calls, SMS) in combination with self-reported data have discovered that there is an association between this kind of activity and changes in bipolar states [37]. However, the use of self-reports such as questionnaires may bring up problems related to self-reporting data compliance. These mental states can be influenced by sleeping patterns, eating habits, social interaction, and physical activity levels [66,67]. Lastly, smartphones can also be used to identify voice features and speaking behavioral data, which can ultimately be used to predict and identify changes in the aforementioned mental states [68]. However, restrictions in phone call features, heterogeneity in patients’ speaking behavior, and voice features can represent a challenge when assessing bipolar disorders.

On the other hand, wearable technologies such as smartwatches or textile embedded sensors can result in a non-invasive way of constantly monitoring patients suffering from bipolar disorder. For example, in [39], a non-invasive wearable platform for acquiring physiological signals is presented, which can be tailored to a single individual providing continuous interaction between clinicians and patients as well as a mobile phone platform for assessment and feedback.

### 5.2. Depressive Disorders

As mentioned before, depressive disorders are characterized by low mood, difficulty to focus, loss of interest or pleasure in activities and diminished self-esteem. These behavioral and physiological characteristics can be tracked using sensing technologies. One of such technologies is Optimal, a sensing platform which provides predictions on patients suffering from depression and stress-related disorders based on the cognitive, motor, and verbal behavior [42], although it lacks data management personalization. Smartphones provide ubiquitous characteristics that can be used in constant monitoring of behavioral patterns related to depressive disorders. Changes in patterns such as sleep habits, phone usage, mobility, and location can be related to depressive symptoms [43].

Finally, smartphones can be used to assess depressive symptoms, a relationship between functional brain activity and the amount of time spent using a phone derived in indicators of depression [69]. In addition to smartphones, wearable devices, such as smart bands or smartwatches, provide an alternative for non-invasive physiological and behavioral on-body tracking. Parameters such as heart rate, body temperature, and galvanic skin response can be used to predict depressive symptoms [44]. There are, however, some device limitations (i.e., battery duration), which can represent a challenge in continuous physiological and behavioral tracking.

### 5.3. Schizophrenia Spectrum Disorder

The use of IoT related technologies to treat and assess patients with schizophrenia and other psychotic disorders proved to be a feasible solution in treating these types of patients. These systems can use smartphones for patients’ daily assessment via questions provided by the professional caregiver. They also provide the doctor with the possibility to visualize the data provided by the patient on a web page or any other mobile device [46]. Still, the adherence to the study was an issue, mainly due to the disorder nature, patients’ condition, and lack in specification on content required from patients.

Changes in digital and social activities, such as phone usage and social interaction can be indicators of relapses in patients suffering from schizophrenia [70], though phone inactivity and misusage and concerns with confidentiality represented challenges when dealing with patients suffering from the aforementioned disorder. In addition to digital and social activities, mobility can be strongly correlated with disease symptoms in patients with schizophrenia [48]. For instance, Crosscheck is a smartphone sensing system that uses a combination of continuous mobile sensing and patients’ self-report to address and predict indicators of schizophrenia levels [47].

### 5.4. Stress-Related Disorders

Stress can affect individuals with long term adverse effects, and in some cases, develop in other mental disorders, such as depression or anxiety [71]. IoT related technologies, such as smartphones, wearables, or ambient sensing devices, provide with new possibilities for addressing and treating stress-related disorders [72]. Physiological characteristics such as voice features can be used to determine levels of stress in individuals. Smartphones’ microphones can be used to collect voice-related data, which have been reported to have a link with individuals’ stress levels [51]; however, devices limitations (i.e., battery duration, phone memory usage) can represent a challenge in continuous voice tracking. In addition, voice features, physical activities, and skin related characteristics, which can be tracked by using smartphones and wearable devices, can be central to dealing with stress-related disorders. For instance, the use of a smart band to detect skin conductance can be used to determine patients’ level of stress [73]. A different type of approach to treating stress-related disorders are the ones being used to treat post-traumatic stress disorder (PTSD), a disorder characterized as a consequence of psychological stress after experiencing a highly stressing event [74]. Sleep patterns, environmental conditions, and physiological features (e.g., heart rate, body temperature) can be used to determine PTSD on individuals. The use of home automation devices and wearable monitoring sensors provides both patients and doctors a nonintrusive method for assessing and treating these disorders [75]. Nevertheless, the use of nonintrusive automated collection data systems presents challenges in terms of security, personal data usage and privacy.

In Table 1, we present a detailed description of each mental disorder addressed by the research community in IoT. We also present how these disorders were currently being reported by the research community in technical areas and relate it to the actual disorders presented in the DSM 5. Also, we point out the types of data collected by each work as well as the IoT technologies used in them, lastly we point out the type of intervention referred in each work. Following Table 1, it can be seen that several of them provide early indications that IoT can be instrumental in dealing with mental issues, either from the patients’, doctors’, or family’s point of view.

In Figure 1, we present the number of studies related to IoT and mental health published per year that derives from this survey. As can be seen, 24 papers have been published since 2010 that relate to IoT and mental health with the majority being related to bipolar disorder as the most studied mental illness. Also, as can be seen in Table 1, smartphones were the most popular technology, which is understandable given the ubiquity and relatively low cost, as per our search criteria.

## 6. Research Challenges in IoT Enabled Mental Health Systems

Research and development of IoT based systems and services for mental health care are increasing, but there are still many challenges that need to be addressed. Although challenges span across different domains, some of these challenges include IoT device performance and the ability to process and analyze massive data to implement intelligent health services [80,81,82]. The possibility to make real-time predictions of critical episodes in patients with mental disorders such as clinical depression also represents a challenge [52]. The psychiatric features or digital biomarkers used to detect and assess any of these mental health-related disorders can be subject to change due to the rapid emergence and acceptance of IoT related technologies, since other measures of mental state and behavior are being developed [83,84]. Even more, some of the symptoms or definitions from the DSM can be unclear or difficult to operationalize [25].

Beyond the developed solutions, other challenges related to the methodology used in studies reviewed in this survey were found. Most of them typically feature a small number of participants for relatively short times due to various reasons such as the vulnerability of the target population, time constraints [42], and the lifespan or validity of emerging technologies, which can be relatively short when compared to clinical trials. Finally, we identified challenges related to security, privacy, and identity management. Some of these challenges refer to the lack of work in these areas and are described in the following subsections.

### 6.1. Data Acquisition for IoT-Enabled Mental Health Systems

The suitability of certain IoT devices for mental health is one of the challenges that demand more attention. That is, the way in which multiple devices are to be used regarding human interaction with their environment to determine their present or future state of mind [85]. Self-quantification can refer to the practices related to finding associations between human health and their surrounding environment by analyzing collected data [86]. Through self-quantification, it is possible to measure and analyze wellbeing-related data from patients’ devices. However, sourcing data in relation to the human environment may require using more than one device which can span wearable, mobile, and devices positioned in different locations across the environments in which patients dwell or move around.

Even more complicated, wearable, mobile, and ubiquitous devices could be from different vendors which at of this moment many of them do not have interoperable data transmission and regulation standards. For instance, there are currently some wellbeing monitoring devices such as Fitbit Charge 3, Apple Watch, Xiaomi Mi band 4, Nike+Fuel and Samsung Gear Fit2 Pro, which encourage users to monitor a wide range of features that can be relevant to mental health patients and professionals such as hours of sleep and personal state of mind. These data are collected via sensors embedded in the device which generate information used for quantification [86,87]. However, in most cases these devices are not directly interoperable, and even if data can be aggregated in a single repository, often data formats vary significantly which makes it hard to analyze. Even more important, some of these devices are not reliable enough to be considered medical grade devices, although this is likely to change [88].

It has been shown that fusing signals or data from heterogeneous devices can pose some issues [85,87,89]. These issues include: reliance on different, non-interoperable protocols; non-standardized data formats; diversity in analysis algorithms and platforms; and variety of data visualization mechanisms. These issues make it difficult for both physicians and patients to produce meaningful clinical decisions due to such fragmentation. The identified issues can affect the usefulness of data acquired for detecting and measuring mental health-related conditions. Perhaps even worse, an individual could own different devices that compute the same data but yield different readings. This can create a major challenge that cannot be easily reconciled as the diagnosis can be contradictory due to surpassing certain established thresholds (e.g., 10,000 steps per day). Also, one aspect to consider is that the data quality regarding precision, reliability, and availability can be a function of usability and patients’ effort in collecting those data. This aspect is not exclusive to mental health, but the consequences of making inferences with incomplete or imprecise data can be critical.

In those situations where patients must use different devices for separate measurement just as mentioned above, the precision and accuracy of these data can become a challenge. A patient may not be able to consistently maintain readings on different platforms for extended periods, as required in mental health cases. Similarly, patients’ perceptions can change over time due to different factors or events, for example, patients with bipolar disorders may suffer manic or depressive episodes which may lead to drastic changes in their mood therefore changing their perception in an extreme way [40,66,67]. However, some studies have shown that the use of patient-family proxy pairs to rate a patient’s quality of life situation can provide with a stable way to assess whether the patient is prone to drastic mood changes [90]. This kind of events can potentially change not only the way an individual thinks and assesses specific situations, but also impact physiological aspects of the individual. If IoT systems are not aware of external events such as those, this can affect the way data quality is assessed which can in turn have effects on mental health automated or semi-automated assessments.

The incorporation of heterogeneous devices represents both an opportunity and a challenge as to determine the number and types of devices as well as types of data that will be adequate for diagnosing each specific mental health condition. This aspect demands particular attention for creating interoperable platforms across varying mental health conditions and enhancing the effectiveness of early detection and treatment of mental illness.

### 6.2. Self-Organization of IoT Devices in Mental Health Systems

As commented earlier, IoT devices can potentially be from different vendors and feature different platforms. This poses another challenge regarding the substitution of devices due to an upgrade or a breakdown. This can be exacerbated if the substituting device provides different capabilities, sensors, readings, precision or frequencies regarding the same data (e.g., socialization, physical activity, sleep). Therefore, self-organization features as such are generally lacking in existing IoT devices and platforms. Self-organization, as observed in nature (e.g., swarms, herds) is desirable and potentially needed for the types of scenarios related to mental health conditions. Seamless coordination in an array of IoT devices and systems is considered a must [91].

An effective diagnosis may require continual monitoring of physiological, behavioral and social data. Therefore, it is important to have devices that can provide compatible or interchangeable data that can be flawlessly integrated into the platform without affecting the system. This will provide robustness and scalability to the platform and, should a device fail, there will be another device to provide comparable data [92]. Then, devices must be able to self-organize themselves with minimal or no intervention from humans. For instance, patients with schizophrenia may be quite inattentive and forgetful due to aspects such as hallucinations and hence leave IoT devices behind or not use the devices as prescribed by the specialists [64,70]. However, recent studies have used self-organized systems to treat patients suffering from post-traumatic stress disorder by combining both home automation and wearable devices [75]. With self-organization systems, IoT devices can announce their presence to other devices in the network and collect specific data as required even if the patient forgets to trigger them.

Self-organization is important to adapt the varying conditions the patient might be encountering. This feature can facilitate monitoring of mental health conditions in and out of the clinic by creating adaptive features to both home and clinic conditions, and this can result in an effective way to aggregate data for timely decisions. Self-organization may provide flexibility among autonomous IoT devices. In this case, independent functional IoT devices monitoring mental health-related symptoms [93]. For example, patients suffering from dementia or cognitive impairment using track and trace technology such as GPS, may get upset if the smartphone they are using have a lesser connection, which can lead to a drop in the use of such technology thus exacerbating the problem [94].

Self-organization can also result in robustness and scalability of IoT devices and platforms as well as increase its functionalities and capacities [92]. Self-organized IoT devices can then collaborate to share resources as they organize themselves to enable energy efficiency, optimal performance, redundant operation -which makes similar devices available if one is down for any reason-, improved quality of service (QoS), and effective and responsive service delivery [95].

### 6.3. Service Level Agreement in IoT Enabled Mental Health Systems

Service Level Agreement (SLA) encapsulates accords where all the terms between a service provider and a service consumer are agreed to maintain the quality of the services [96]. All types of IoT devices must ensure the trust, privacy, ethics, transparency, and security of a service consumer. This assurance is applied to data throughout the platform architecture ranging from sensor data, aggregation, and sharing mechanisms, third parties, storage, and also within transit.

Explicit consent must be taken from the patient for collecting, sharing, allowing analytics, removing data, storing data, and the like. To tackle this challenge the European Union introduced the General Data Protection Regulation (GDPR) [97]. Other data protection laws are also being developed in other regions such as in India and Brazil [98]. Simultaneously, the Association for Computing Machinery (ACM) has just presented an initiative to create the ACM Technology Policy Council (“ACM Global Technology Policy Council”, n.d.) aimed at opening the conversation related to the impacts of information technology and computing on public policies across the globe.

Networks and information systems directives have been formulated to protect the network that would carry such data [99]. To achieve a standard in IoT healthcare globally, countries should also initiate the formulation of data and network regulation and laws. Proper legal consent should be in place in case of self-organized IoT devices, even if legislations vary across regions for transnational companies span across different regions and countries. This issue takes an especial relevance when patients suffering from a given mental disorder may not have a clear judgement. This is particularly relevant when some patients may consent to sharing certain data, but different circumstances may change their minds or even worse may not recall having taken the decision of sharing data. To illustrate, a patient suffering from Alzheimer’s disease may forget about the consent she or he previously provided [94]. However, chatbots and conversational agents have been used to treat patients with mental disorders such as dementia, Alzheimer’s and social isolation, providing them with an empathetic and emotional replacement for a human interlocutor [29,30,76]. In such a situation, a legally appointed representative may provide consent. This raises concern as legal consent should be in place for the individual who is providing consent on behalf of the patient.

Terms and conditions between the device manufacturer and the healthcare service provider should be clear, for example, any digital certification on the device, computing power and energy consumption of the device. Proper terms should also be defined in the SLA when a third-party software or service runs on the device. SLA for these devices should precisely define the types of data it will extract from the patient and how it will be processed. Acceptable risk related to unreliable data produced through the IoT devices must be considered, and proper terms should be incorporated in the SLA. Terms should be defined when two IoT device devices interact with others as to what is the nature of the interaction and the process by which it will take place. Terms should be well crafted in the SLA when a device manufactured in a particular geographic location moves to another geographic location. For example, GDPR would apply in the European Union but should patients travel to India or any other country, a different set of laws would apply and the negotiable and re-negotiable terms should be defined for the proper working of the device. This is particularly relevant in regions where work migration is relatively high, such as between Mexico and the United States.

SLA becomes important when it comes to personal data being shared across the Internet. In countries like Panama, the data storage clause is entirely missing in their data protection law. The wide range of rich and abundant behavioral, physiological and social data being collected from people’s health will be at risk if countries around the world do not sign international accords regarding a common data protection law or even just have proper data protection law in place.

### 6.4. Identity Management

IoT based solutions for supporting individuals with mental health issues may: collect sensitive/critical information [16,100]; communicate patient data over the Internet [11]; perform uninterrupted monitoring and detection of critical illness [101], and enable the collection of targeted records [102]. However, security remains a core challenge and a leading concern in all IoT enabled research, particularly notable in solutions which are focused on mental health conditions [103,104]. Although IoT provides numerous benefits, it also presents new risks, vulnerabilities, and other security challenges that may harm a patient’s safety and health, such as loss of medical histories and prescriptions, unauthorized access to devices, and identity theft [105,106,107]. A fundamental approach to overcoming these security challenges depends on having a robust and secure identity management system. To secure sensitive and critical information of patients, IoT devices, patients, caregivers, healthcare providers, health insurances, pharmacies or drug stores, and related institutions need to be uniquely identified, verified, and authenticated to establish secure communications before they are granted any access to the patient’s data [107].

Existing identification and authentication systems are centralized, untrusted, non-scalable, and represent a single point of failure/compromise [108,109]. In addition, these solutions were conceived in a pre-IoT era and therefore are incompatible with requirements inherent to IoT such as its decentralized topology, and resource-constrained devices. Existing identity management systems are facing challenges when implemented in IoT systems due to multiple reasons. First, existing identity management systems have several limitations such as ineffective management of billions of devices, susceptibility to various attacks, limited scalability, siloed identity management and removal of the locus of identity control from the identities owner [110,111,112]. IoT devices used in healthcare must transfer patient data over the network with little or no intervention from humans. Thus, they need to be secured from attacks by implementing effective identity management [113,114,115]. Second, when unknown and unmanaged devices interact with the IoT network they can lead to new security vulnerabilities and additional risks [116]. Finally, conventional identity management systems are not necessarily compatible with IoT identity management since they are not designed to handle tens of billions of identities, have an inflexible architecture, are proprietary standard oriented, difficult to use, and less adaptive to various scenarios [117]. IoT devices and services to be implemented for mental health need to ensure secure, persistent, scalable, movable and interoperable new identity management systems which cannot be compromised due to legacy networks.

Verifiable ownership of IoT devices and their identities is a core challenge of the current identity management systems. The challenge makes it even worse when it comes to IoT devices involved in mental health which store and transmit sensitive information. Existing systems and healthcare service providers solely control the devices and all other identities exposing them to various attacks. For example, patients suffering from schizophrenia presented concerns regarding confidentiality and the potential consequences that the use of information provided could be used against them by clinicians, employers or insurance providers [46]. However, the use of patient-family proxy pairs may be a solution in situations where a patient presents confidentiality distrust, giving a close familiar the possibility to make a decision on their behalf [90,118]. Thus, the identities of a patient, devices, and participants in the mental health domain need to be in a decentralized environment to ensure the full control and ownership of devices and corresponding identities. This enables tracking of any activities performed on a patient’s mental health system and ensures that any access happens with the consent of those who are responsible and verified.

## 7. Conclusions and Future Perspectives

Mental health disorders are becoming a global issue. For instance, due to the current COVID-19 pandemic, social isolation and stress-related situations have become more common, thus leading to mental health tolls in the majority of people who are carrying their day-to-day tasks from home [119,120]. In addition to this, people who are concerned about their future and their families, specifically those for whom the pandemic resulted in the loss of jobs or at worst, the loss of a close tie, are prone to suffer mental health-related problems, such as anxiety, depression, or stress-related conditions. Also, medical staff constantly exposed to a considerable degree of stress, anxiety, depression, and insomnia due to physical fatigue, work burnout, and day-to-day tasks that cannot be carried out from home are not exempted from these mental health tolls [121]. New ways of detecting and potentially treating mental disorders are being proposed and developed. For instance, the use of social robots can help reduce the mental fatigue of those people who are in social isolation due to the pandemic. Additionally, the use of telerobots can provide the ability to measure a patient’s temperature without any physical interaction [76,122]. However, the potential of IoT for mental health applications has been hardly harnessed. As for future development, we next present some challenges that we think need to be addressed to facilitate future research. First, sensing technologies must be able to make inferences about critical episodes. For instance, patients suffering from major depressive disorders may have suicidal thoughts, which may lead to potential death. Also, devices must be able to adapt themselves to circumstances with minimal or no intervention from researchers or technicians (e.g., patients with schizophrenia may be quite inattentive and forgetful due to aspects such as hallucinations and hence not use the devices as prescribed). Secondly, patients’ perceptions can change over time due to illness, treatment side effects or other events, which can interfere in the patient-technology-interaction (e.g., patients with bipolar disorders may suffer manic or depressive episodes leading to drastic changes in self-assessment or perception of reality). Lastly, intervention wise, several challenges are yet to be discussed in terms of policies, laws, and research protocols. As a prominent example, some of the symptoms or definitions presented in the DSM 5 can be unclear or difficult to operationalize, thus resulting in overlaps between symptoms and diagnoses [25]. In this work, we presented an overview of the main research contributions from IoT in the mental health area. We particularly focused on identifying previous works and related open for implementing such technological solutions, ranging from data acquisition and self-organization of devices to security and privacy.

There are plenty of opportunities for IoT to play a key role in boosting and democratizing mental health care, yet we stand at a very early stage of this new paradigm. IoT devices are characterized by generating an enormous amount of data, which can be used to diagnose mental health related cases, but unfortunately these data are scattered around on platforms that are still vendor dependent. This makes interoperability a major issue in the identified mental-health related solutions discussed above. The diversity of devices also pose important issues with respect to their widespread and seamless use, thus making the self-organization of the IoT ecosystem of paramount relevance to improve the robustness and scalability of solutions designed for mental health and wellbeing.

Like for any other solution, security turns to be of major importance in IoT for mental health. An efficient identify management system is the foundation in securing the IoT-based mental health solutions to protect the overall ecosystem and sensitive information of patients and other stakeholders. The proper implementation of these mechanisms could empower full ownership and control for identity holders over their identities with transparency, portability, and easy management.

## Figures and Tables

**Figure 1 ijerph-18-01327-f001:**
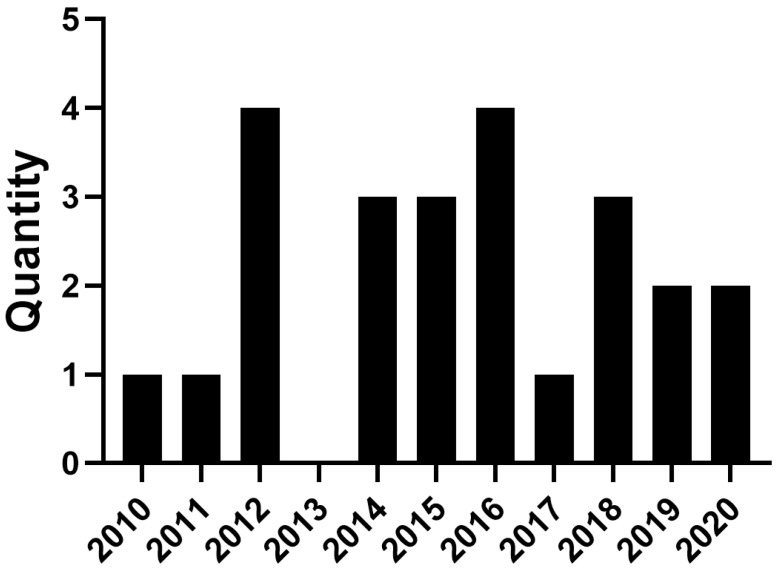
Number of studies related to IoT and mental health published per year.

**Table 1 ijerph-18-01327-t001:** Studies in IoT-enabled systems for mental health.

DSM 5 Disorder	Reference	Reported Mental Disorder(s)	Measures			Technologies			Approach		Challenges			
			Physiological Data	Behavioral Data	Social Data	Wearable	Smartphone	Embedded	Detection/ Diagnosis	Treatment	Data Acquisition	Self- Organization	Service Level Agreement	Identity Management
Trauma and stressor- related disorder (5)	[72]	Psychological Stress	✓	✓		✓	✓			✓	✓			
	[51]	Mental stress	✓				✓		✓	✓	✓			
	[75]	Post-traumatic stress disorder	✓	✓		✓		✓		✓	✓	✓		
	[73]	Mental Stress	✓	✓		✓	✓		✓	✓	✓			
	[76]	Mental Stress			✓		✓	✓		✓				
Depressive disorders (5)	[42]	Depression and stress- related disorders	✓	✓		✓	✓		✓		✓			
	[44]	Depression	✓	✓	✓	✓	✓		✓		✓			
	[43]	Depression		✓			✓		✓		✓			
	[69]	Depression	✓	✓			✓		✓	✓	✓			
	[77]	Depression		✓			✓		✓		✓			
Bipolar and related disorders (1)	[40]	Bipolar disorder	✓	✓		✓			✓		✓			
Bipolar I disorder (6)	[52]	Bipolar disorder (manic-depression psychosis)	✓	✓	✓		✓			✓	✓	✓		
	[38]	Bipolar disorder (manic-depressive disorder)		✓	✓		✓			✓	✓			
	[37]	Bipolar disorder	✓	✓	✓		✓		✓		✓		✓	✓
	[68]	Bipolar disorder (manic-depressive disorder)	✓	✓	✓		✓		✓		✓			
	[67]	Bipolar disorder (manic-depressive disorder)		✓			✓		✓		✓			
	[66]	Bipolar disorder		✓	✓		✓		✓		✓			
Bipolar II disorder (1)	[39]	Bipolar disorders (Depression, hypomania, mixed state, and euthymia)	✓	✓		✓	✓			✓	✓	✓		✓
Schizophrenia spectrum disorder (6)	[46]	Schizophrenia		✓	✓		✓			✓	✓			
	[70]	Schizophrenia		✓	✓		✓			✓	✓			
	[48]	Schizophrenia		✓	✓		✓		✓	✓	✓			
	[47]	Schizophrenia		✓	✓		✓		✓	✓	✓			
	[78]	Schizophrenia	✓	✓			✓		✓	✓	✓			
	[79]	Schizophrenia	✓	✓	✓		✓	✓	✓		✓			
Total	24		14	22	12	7	22	3	16	11	23	3	1	2

## Data Availability

Not applicable.

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
