# Peer review of "Internet of Things for Mental Health: Open Issues in Data Acquisition, Self-Organization, Service Level Agreement, and Identity Management"

_ijerph, 2021, doi:10.3390/ijerph18031327_

Round 1

Reviewer 1 Report

The manuscript presents scientific mainly papers and studies in an organized manner with potential open issues for the effective use of IoT systems in mental health.

The manuscript is well organized and the presentation is clear.

There are lots of issues that are of interest in apps and IoT for mental health patients such as data acquisition issues, lack of self-organization of devices and service level agreement, and security, privacy and consent issues, among others.

The manuscript can be strengthened if it presents for each and one of the above issues which solutions are covering them adequately, and which fail to meet the each one of the above requirements.

Table A1 shall be expanded to include multiple columns of challenges to show all the requirements and challenges that are or are not covered.

Futhermore, there should be a critical discussion on the proposed best or better solutions that are included for each desease.

Reviewer 2 Report

This paper presents a comprehensively survey on Internet of Things for mental health, which introduces the most prevalent mental health disorders, presents the IoT technologies that could be used to sense the behavioral, physiological and social signal data, and presents current state of the art in IoT based mental health systems.

Overall the paper is well written, but can be made better in the following points:

  1. What is the difference between your review paper and other review papers in this field? It seems not to compare from this perspective.
  2. Your literature is relatively sufficient, expounding the application of the Internet of Things in mental health in detail, but if you add some of your own opinions, it will be more perfect.
  3. Page 16-Page 28 are full of charts, it feels like a waste of space, and not well organized, so it would be better if it can be clearly expressed.

Minor points:

-The 5th line , This paper comprehensively surveys works... -> This paper comprehensively survey works...

-The 338th line, but also but also impact... -> but also impact...

(Check for other grammatical errors, and please correct them.)

Reviewer 3 Report

The authors carry out a very interesting study about the increase of mental illness cases around the world and how technological paradigms such as the Internet of Things (IoT) provide us with new capabilities to detect, assess, and care for patients early.

In view of the effects that the current pandemic has shown on people's mental health (e.g. due to social isolation), I would recommend the authors to devote some text in the manuscript to discuss about how IoT could be used to help deal with such effects.

Moreover, I would suggest the authors to describe in more detail the work they anticipate should be considered for the near future in order to advance this field.

Round 2

Reviewer 2 Report

The author fully revised the paper in accordance with the previous revision comments and completed it well. After reading the revised paper, there are some suggestions I hope the author will refine it:

1. Can you comprehensively evaluate other methods and explain the future development direction of the industry.? Can you give suggestions for future development to facilitate future research.

2. Figure 1 needs to be adjusted again. Its expression is not intuitive enough. Its y-axis should be quantity instead of frequency. Moreover, the author needs to explain the meaning of the picture in more detail, this picture is a bit difficult to understand.

3. Table 1 needs to adjust the layout.
